

# The use of four-pillar regimen for heart failure management: results from the Jordanian Heart Failure Registry (JoHFR)

Mahmoud Izraiq[1], Mustafa Jammal[2], Ahmad A. Toubasi[3], Sae'ed Mari[1], Sarah AlNajafi[1], Ayad Al-Qadasi[1], Khaled Al Maharmeh[1], Maha Almansour[1], Soadad Saleh[1], Yaman Ahmed[4] and Hadi Abu-hantash[5]

[1] Specialty Hospital, Amman, Jordan
[2] Al-Hayat Hospital, Amman, Jordan
[3] School of Medicine, University of Jordan, Amman, Jordan
[4] King Abdullah University Hospital, Irbid, Jordan
[5] Amman Surgical Hospital, Amman, Jordan

## ABSTRACT

**Background**. Heart failure (HF) is a complex cardiovascular disease. Effective management typically involves four main medications: angiotensin-converting enzyme inhibitors/angiotensin receptor blockers, beta-blockers, angiotensin receptor-neprilysin inhibitors, and mineralocorticoid receptor antagonists, along with sodium-glucose co-transporter-2 inhibitors (SGLT2i). The primary objective of this article is to assess and identify the utilization of four-pillar regimen for HF managment and explore the characteristics of the patients being on the four-pillar regimen in Jordan.

**Methods**. Data from the Jordanian HF registry (JoHFR) was analyzed, encompassing records of HF patients treated in various cardiology centers from 2021 to 2023.

**Results**. The medical records of 2,151 patients with HF who were admitted to cardiology centers throughout Jordan were collected. Males comprised 58.0% of the included patients. Moreover, 71.0% of patients had chronic HF, whereas the rest, 29.0%, had acute HF. Of these, only 0.6% received the complete four-pillar treatment of HF. Beta-blockers were the most frequently used medication, prescribed to 74% of patients, while SGLT2i were the least common, used by only 9%. Notably, patients with type 2 diabetes were more likely to be on the four-pillar regimen ($P$-value = 0.016). Additionally, patients with a glomerular filtration rate (GFR) below 60 were more likely to be treated using the four-pillar ($P$-values = 0.044). The analysis revealed no significant difference in mortality rates between the two groups ($P$-value = 0.475).

**Conclusion**. Our study demonstrated an overall low utilization of the four-pillar regimen for HF treatment in Jordan with several patients' characteristics associated with it. This highlight the need for enhanced collaborative effort and governmental initiatives to address the challenges of the low utilization of these medications.

Corresponding authors
Mahmoud Izraiq, izraiq@yahoo.com
Ahmad A. Toubasi,
tubasi_ahmad@yahoo.com

## INTRODUCTION

Heart failure (HF) is a complicated and debilitating disease of the cardiovascular system characterized by the inability of the heart to function normally and to pump blood properly and adequately, leading to impairment of cardiac input (*Beezer et al., 2022*). As one of the major causes of morbidity and mortality, HF places a significant burden on health systems worldwide, including Jordan (*Bhagat et al., 2019*). As the prevalence of HF is on the rise, the importance of in-depth investigation and assesing of the current guideline-directed medical therapy is essential, along with how adherent the physicians and the system are in following it. This is fundamental towards improving and identifying factors associated with compliance to these guidelines (*Bhagat et al., 2019*).

The main intervention for successfully treating HF is pharmacological (*Beezer et al., 2022*). The main drugs used worldwide in the management and treatment of HF are angiotensin-converting enzyme inhibitors (ACEi), beta-blockers (BBs), angiotensin receptor blockers (ARBs), angiotensin receptor-neprilysin inhibitors (ARNIs), mineralocorticoid receptor antagonists (MRAs), and sodium-glucose co-transporter-2 inhibitors (SGLT2i) (*Beezer et al., 2022*). Among patients with HF with reduced ejection fraction (HFrEF) the guidelines are more clear relative to patients with HF with preserved ejection fraction (HFpEF). ACEi, ARNI, BBs, and SGLT2i are considered the main disease-modifying treatments for improving symptoms, reducing hospital admissions, and increasing survival (*Tomasoni et al., 2019*; *McMurray & Pfeffer, 2005*). However, only SGLT-2i and MRAs improves mortality among patients with HFpEF (*Mosterd & Hoes, 2007*).

The primary objective of this article is to assess and identify the utilization of four-pillar regimen for HF managment and explore the characteristics of the patients being on the four-pillar regimen in Jordan.

## METHODS

### Setting, design, and population

We analyzed data from the ongoing Jordanian HF Registry (JoHFR). This is a national HF registry in Jordan. Detailed methodology information is found in the protocol registration (NCT04829591). In brief, medical records of patients with HF who visited cardiology medical centers across Jordan from 2021 to 2023 were reviewed. The study population included patients who were seen in the inpatients wards or outpatient clinics with a diagnosis of HF. A total of 21 centers were involved in the study, including private centers, teaching and nonteaching hospitals, and tertiary university hospitals. Patients were divided into two groups: patients on the four-pillar regimen and patients who are not on it. In this prospective multicenter registry, patients are being followed up to 1 year after the first medical record was collected. Patients under the age of 18 were excluded from the study. According to Jordanian research data collection guidelines, each site's principal investigator was responsible for getting an institutional review board (IRB) approval from their institution. The IRB of the Specialty Hospital approved the conductance of this research (approval number: 5/1/t/104826). Patients were informed of the conduct of a HF

registry and given the option to reject their data entry. A written informed consent was obtained from all the participants. All patients' data were dealt with anonymously.

## Data collection

An online Google form was distributed among the data collection team. This form was based on similar forms made by other heart failure registries. In summary, the form had several sections, including demographic characteristics of the patients, baseline co-morbidities, baseline laboratory investigations, what medications the patients were on, and clinical outcomes including 30-day mortality, need for mechanical ventilation, and length of hospital stay.

It was then shared into a database managed by this research's administrative and analytic team. HFrEF was defined by an echocardiogram ejection fraction <40%; meanwhile, HFpEF is diagnosed by an echocardiogram with an ejection fraction of >50% and a diastolic dysfunction.

In 2021, the American College of Cardiology (ACC) and the American Heart Association (AHA) gave SGLT2i a class 1 indication for treating HFrEF and a class IIa indication for both mildly reduced EF (41–49%) and HFpEF ($\geq$50%). These recommendations made SGLT2i one of the main four pillar regimen for treating HF. Accordingly, the main four-pillar regimen for treating HF in this study were BB, ACEI/ARBs, ARNI, and SGLT2i.

## Data analysis

The data was entered using Microsoft Office Excel 2019 and then imported and analyzed using IBM SPSS v.25 software (IBM, Armonk, NY, USA). Descriptive statistics of mean and standard deviation for continuous variable and frequency with percentages for categorical variables were presented. To detect significant mean differences between the study groups for continuous variables, student $t$-tests were applied. Chi-squared tests were applied for categorical variables. We have conducted survival analyses using Kaplan–Meier curve to assess the difference between patients who were on the four-pillar regimen for HF and patients who were not. A $p$-value < 0.050 was considered statistically significant.

## RESULTS

### Characteristics of the included patients

The medical records of 2,151 patients with HF who were admitted to cardiology centers throughout Jordan were collected. Males comprised 58.0% of the included patients. Furthermore, 69.2% of the patients had type 2 diabetes, whereas 80.7% had hypertension. The percentage of smoking and alcohol among the included patients was 31.3% and 0.6%, respectively.

Moreover, 71.0% of patients had chronic HF, whereas the rest 29.0% had acute HF. A percentage of 70.3% of patients had HFrEF. whereas, 29.7% had HFpEF. Patients who were on the four-pillar of HF were 0.6% of all study participants. Figure 1 describes the use of each of the four-pillar in HF management. The most commonly used medication was BBs (74%), while the least commonly used were ARNI (10%) and SGLT2i (9%). Figure 2 shows the percentage of usage of the four-pillar regimen for HF among the included patients.

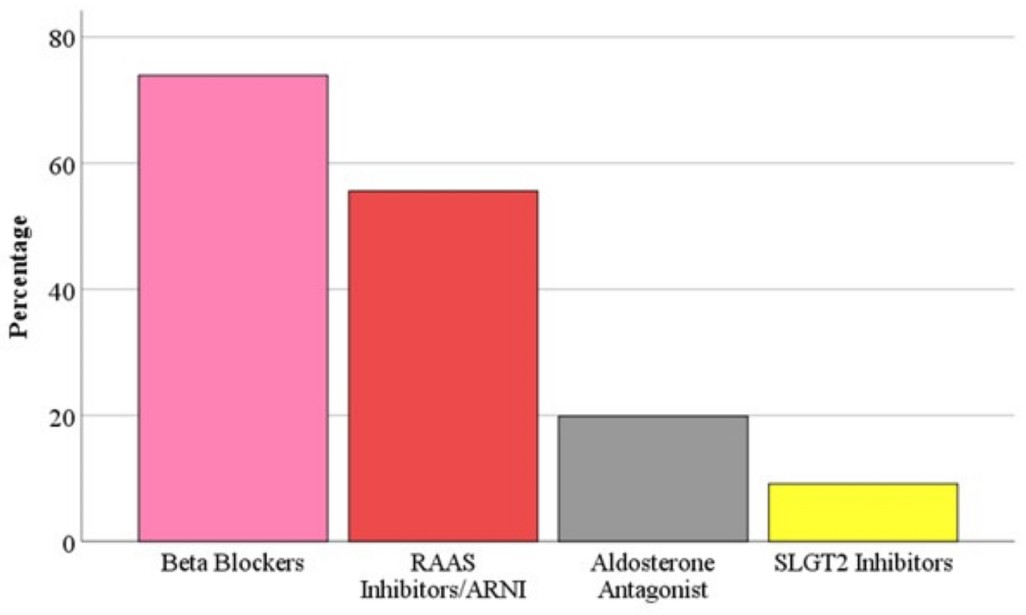

**Figure 1** The usage of each class of the four-pillar of HF management.

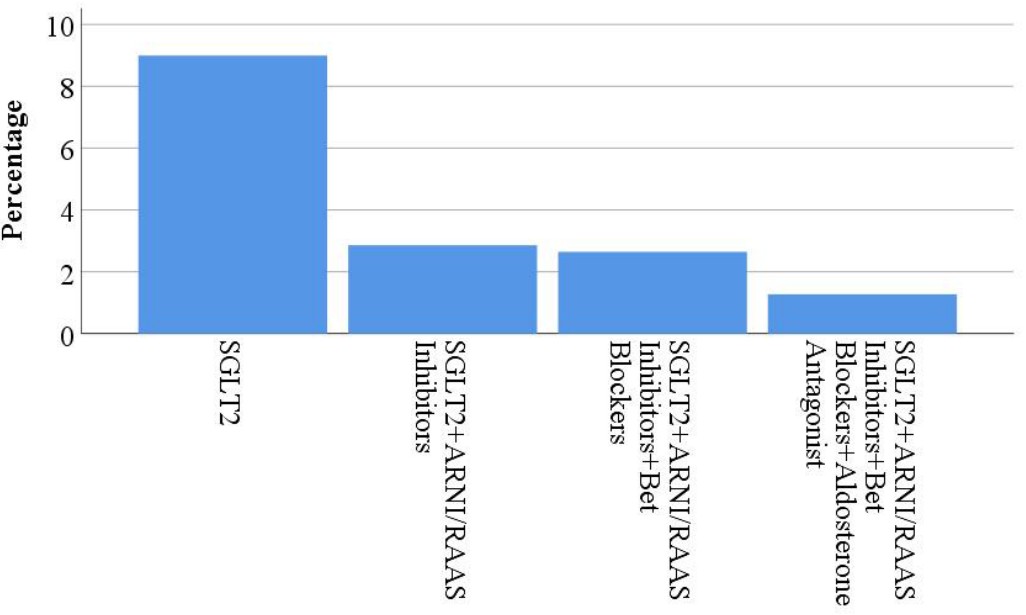

**Figure 2** The usage of the four-pillars of HF management.

## Comparison of the characteristics of the patients according to the use of the HF four-pillar regimen

Table 1 summarizes the variable studied between the two study groups: on four-pillar regimen for HF management and not on the regimen. A pronounced discrepancy was

**Table 1 The difference between patients who were on the four-pillar regimen for HF and who were not.**

| Variable | | Patients on the four-pillar regimen for HF ($n = 13$) | Patients not on the four-pillar regimen for HF ($n = 2,138$) | *P*-value |
|---|---|---|---|---|
| Heart failure | Acute | 5 (38.5) | 608 (28.9) | 0.797 |
| | Chronic | 8 (61.5) | 1,494 (71.1) | |
| Sex | Male | 8 (61.5) | 1,227 (58.0) | 0.797 |
| | Female | 5 (38.5) | 888 (42.0) | |
| Age | <40 | 0 (0.0) | 87 (4.4) | |
| | 40–49 | 0 (0.0) | 142 (7.2) | |
| | 50–59 | 0 (0.0) | 327 (16.6) | 0.536 |
| | 60–69 | 3 (37.5) | 519 (26.3) | |
| | ≥70 | 5 (62.5) | 900 (45.6) | |
| Hypertension | Yes | 11 (84.6) | 1,604 (80.7) | 0.720 |
| | No | 2 (15.4) | 384 (19.3) | |
| Diabetes | No | 0 (0.0) | 619 (31.0) | 0.016[*] |
| | Yes | 13 (100.0) | 1,375 (69.0) | |
| Smoking | No | 10 (76.9) | 1,364 (68.6) | 0.520 |
| | Yes | 3 (23.1) | 624 (31.4) | |
| Alcohol | No | 13 (100.0) | 1,976 (99.4) | 0.779 |
| | Yes | 0 (0.0) | 12 (0.6) | |
| Dyslipidemia | No | 3 (23.1) | 840 (42.3) | 0.163 |
| | Yes | 10 (76.9) | 1,148 (57.7) | |
| Obesity | No | 12 (92.3) | 1,828 (92) | 0.962 |
| | Yes | 1 (7.7) | 160 (8.0) | |
| Positive family history of ASCVD | No | 13 (100.0) | 1,881 (94.6) | 0.390 |
| | Yes | 0 (0.0) | 107 (5.4) | |
| Positive family history of premature death | No | 11 (84.6) | 1,529 (76.9) | 0.511 |
| | Yes | 2 (15.4) | 459 (23.1) | |
| History of ASCVD | No | 1 (12.5) | 308 (19.5) | 0.619 |
| | Yes | 7 (87.5) | 1,274 (80.5) | |
| History of implanted device | No | 7 (87.5) | 1,519 (96.0) | 0.221 |
| | Yes | 1 (12.5) | 63 (4.0) | |
| Known history of HF prior to this admission | No | 1 (7.7) | 396 (18.7) | 0.309 |
| | Yes | 12 (92.3) | 1,721 (81.3) | |
| Number of admissions and office visits in the past 6 months | 0 | 2 (18.2) | 852 (50.8) | |
| | 1 | 4 (36.3) | 409 (24.4) | |
| | 2 | 2 (18.2) | 161 (9.6) | 0.275 |
| | >2 | 3 (27.3) | 254 (15.2) | |
| Cholesterol (milligram/deciliter) | Normal | 5 (100.0) | 676 (85.7) | 0.361 |
| | High (>200) | 0 (0.0) | 113 (14.3) | |

| Variable | | Patients on the four-pillar regimen for HF (*n* = 13) | Patients not on the four-pillar regimen for HF (*n* = 2, 138) | *P*-value |
|---|---|---|---|---|
| LDL (milligram/deciliter) | Normal | 5 (100.0) | 699 (87.4) | 0.396 |
| | High (>130) | 0 (0.0) | 101 (12.6) | |
| HDL (milligram/deciliter) | Normal | 1 (20.0) | 218 (27.8) | 0.698 |
| | Low (<40) | 4 (80.0) | 566 (72.2) | |
| Triglycerides (milligram/deciliter) | Normal | 2 (40.0) | 502 (63.4) | 0.280 |
| | High (>150) | 3 (60.0) | 290 (36.6) | |
| BNP | Normal | 0 (0.0) | 24 (3.3) | 0.749 |
| | High | 3 (100.0) | 703 (96.7) | |
| NT-ProBNP | Normal | 0 (0.0) | 30 (3.2) | 0.582 |
| | High | 3 (100.0) | 296 (90.8) | |
| Sodium (milligram/deciliter) | <136 | 7 (53.8) | 600 (29.9) | 0.153 |
| | 136–145 | 6 (46.2) | 1335 (66.5) | |
| | >145 | 0 (0.0) | 74 (3.7) | |
| Potassium (milligram/deciliter) | <3.5 | 0 (0.0) | 108 (5.4) | 0.680 |
| | 3.5–5 | 11 (84.6) | 1636 (81.6) | |
| | >5 | 2 (15.4) | 261 (13.0) | |
| Hemoglobin (milligram/deciliter) | ≥10 | 12 (100.0) | 1,580 (83.4) | 0.123 |
| | <10 | 0 (0.0) | 314 (16.6) | |
| eGFR | ≥60 | 0 (0.0) | 373 (57.7) | 0.044[*] |
| | <60 | 3 (100.0) | 274 (42.3) | |
| BUN (milligram/deciliter) | Normal | 1 (11.1) | 438 (24.2) | 0.360 |
| | >20 | 8 (88.9) | 1373 (75.8) | |
| HBA1c (milligram/deciliter) | Normal | 0 (0.0) | 320 (36.1) | 0.066 |
| | >6 | 6 (100.0) | 567 (63.9) | |
| Ejection fraction | ≥50 | 4 (36.4) | 549 (29.7) | 0.628 |
| | <50 | 7 (63.6) | 1301 (70.3) | |
| Creatinine (micromoles/L) | Normal | 7 (58.3) | 1,113 (57.5) | 0.951 |
| | >115 | 5 (41.7) | 824 (42.5) | |
| 30 days mortality | No | 11 (84.6) | 1,934 (90.5) | 0.475 |
| | Yes | 2 (15.4) | 204 (9.5) | |
| Mechanical ventilation | No | 6 (85.7) | 1,302 (95.3) | 0.233 |
| | Yes | 1 (14.3) | 64 (4.7) | |
| Length of hospital stay | | 5.00 ± 2.20 | 6.25 ± 7.63 | 0.644 |

**Notes.**

ASCVD, atherosclerotic cardiovascular disease; BUN, bilirubin urea nitrogen; BNP, b-type natriuretic peptide; eGFR, estimated glomerular filtration rate; HF, heart failure; HDL, high density lipoprotein; HBA1c, glycated hemoglobin; LDL, low density lipoprotein.

*$p < 0.050$.

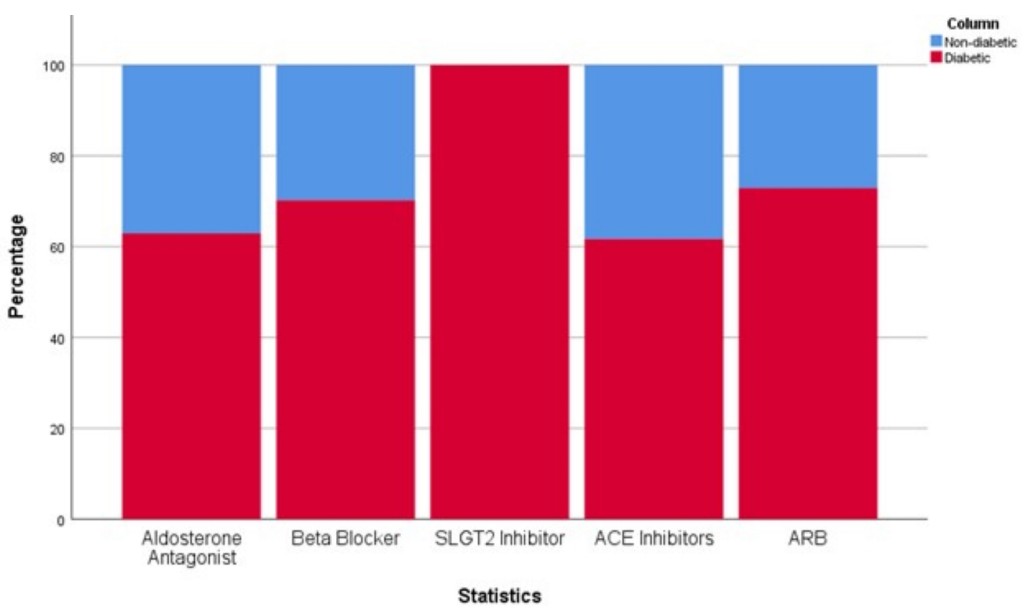

**Figure 3 Distribution of the use of the four-pillar of management of HF across diabetes status.**

observed in the prevalence of type 2 diabetes between these groups. Remarkably, all patients with type 2 diabetes were treated with the four-pillar regimen (*P*-value = 0.016). This association is further visualized in Fig. 3, which delineates the proportional use of each of the four-pillar across patients with and without type 2 diabetes, demonstrating a higher utilization rate among individuals with type 2 diabetes. Furthermore, every patient prescribed SGLT2i had a concurrent diagnosis of type 2 diabetes.

The employment of the four-pillar treatment strategy was also evaluated across different types of HF. Figure 4 highlights a greater percentage of patients with HFrEF receiving all four medications in contrast to those with HFpEF. The analysis indicated no significant distinction in the use of the four-pillar when comparing patients with acute HF to those with chronic HF. Also, there was no significant difference in the use of the four-pillar between sex, or across the variables of hypertension , smoking, alcohol use, dyslipidemia, obesity, or family history of atherosclerotic cardiovascular disease or premature death.

All patients who had GFR<60 milliliter/minute or glycated hemoglobin (HbA1c)>6.0% were found to be on four-pillar regimen (*P*-value = 0.044, and 0.066, respectively). In contrast, no significant differences were observed between the two groups in other laboratory findings, such as lipid profiles, B-type natriuretic peptide, N-terminal pro b-type natriuretic peptide, sodium, potassium, hemoglobin, urea, and creatinine levels. The 30-day mortality rate did not significantly differ between the study groups (*P*-value = 0.475).

Survival analysis demonstrated that patients who were not on the four-pillar regimen for HF had worse survival compared to their counterparts. However, the difference was not statistically significant (HR = 1.954; 95% CI [0.272–14.010]) (Fig. 5).

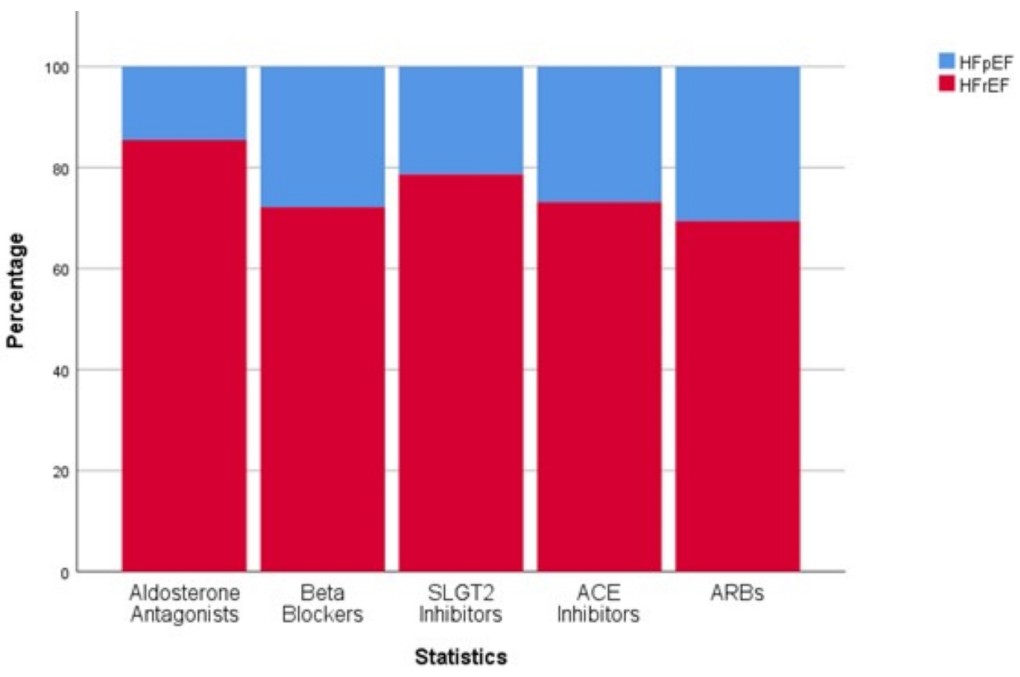

**Figure 4  Distribution of the use of the four-pillar of management of HF across HF types.**

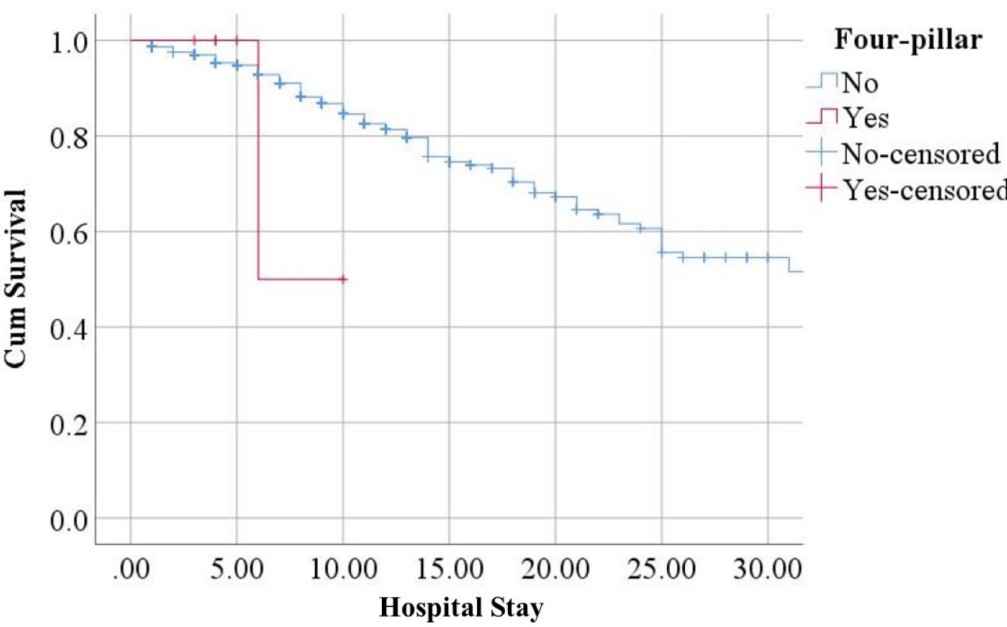

**Figure 5  Survival analysis.**

## DISCUSSION

This is the first large-scale national prospective multi-center registry to study patients with HF in Jordan. Our study revealed that only 0.6% of patients with HF were prescribed the optimal combination therapy. This low percentage of utilizing these medications can be attributed to several factors, including cost and availability. SGLT2i are relatively newer agents demonstrating substantial cardiovascular benefits in patients with HF. However, their higher acquisition costs compared to traditional HF medications, coupled with their limited availability within healthcare facilities, may contribute to the lower prescription rates. Cost-related factors and medication availability also play a crucial role in the utilization of other HF medications. Patients with limited financial resources or inadequate health insurance coverage may face barriers to accessing prescribed medications, leading to non-compliance or suboptimal treatment.

Similarly, the availability of certain medications within healthcare facilities, particularly in resource-limited settings, can influence prescription patterns. Addressing these barriers through policy interventions, such as improved reimbursement systems or the inclusion of essential medications in formularies, may help increase the utilization of guideline-recommended HF therapies. The suboptimal usage of combined medications raises concerns about the potential impact on patient outcomes. Several studies have demonstrated the significant benefits of these medications in improving survival, reducing hospitalizations, and enhancing quality of life in HF patients (*Aguilar et al., 2009*; *Hassanein et al., 2015*). Cardiovascular risk factors and co-morbidities, as well as cost and availability, are several patient-related factors that impact the selection and usage of HF medications.

Our study highlights that there was an association between the use of the four-pillar of HF and comorbid conditions such as type 2 diabetes mellitus, and kidney function. These factors are found to be significant in influencing medication choices and dosing strategies. In a long-term HF registry done by the European Society of Cardiology, and included patients with chronic HF, the prevalence of type 2 diabetes, atrial fibrillation, and renal dysfunction were 31.9%, 37.7%, and 18.4, respectively (*Crespo-Leiro et al., 2016*). HF itself is considered an insulin-resistant state and is associated with a significant risk for the future development of type 2 diabetes (*Aguilar et al., 2009*). Higher HbA1C levels are associated with increased mortality, which is mainly multifactorial and may involve both direct and indirect effects of hyperglycemia. Adverse effects of hyperglycemia are potentially represented by endothelial dysfunction, high oxidative stress, increased protein kinase C activation, and potentially accelerated atherosclerosis (*Aguilar et al., 2009*). In addition, the magnification of glycation end products (AGEs), resulting from chronic hyperglycemia, may lead to many detrimental processes such as increased myocardial stiffness and upregulation of cellular signals that results in cellular dysfunction (*Aguilar et al., 2009*). Also, elevated levels of HbA1C may be an indicator for a greater level of insulin resistance that is associated with derangements of myocardial energy and cardiac metabolism and increased activation of the sympathetic nervous system (*Aguilar et al., 2009*). Finally, high HbA1C levels also may be reflective of poor compliance with medications, which in turn may be associated with poor outcomes, and that could clarify the considerable association between high HbA1C and the significant

percentage of patients with HF who were on the four-pillar regimen (*Beezer et al., 2022*). In addition, 100% of the patients on SGLT2i had type 2 diabetes. This highlights that SGLT2i is still being considered as a medication to control HbA1c and type 2 diabetes and not a main pillar of heart failure treatment as recommended by the international guidelines (*Tomasoni et al., 2019*). Similarly, impaired renal function was also associated with the physicians' adherence with prescribing the four-pillar regimen for HF.

Moreover, our study included patients with HFpEF and HFrEF and demonstrated that the prevalence of the use of the four-pillar was double among patients with HFrEF compared to patients with HFpEF. This result highlight that patients with HFrEF were more likely to be on the four-pillar which is consistent with the most recent European and American guidelines in the management of HF (*McDonagh et al., 2021*; *Heidenreich et al., 2022*). Our findings also highlight that a percentage of patients with HFpEF also are on the four-pillar regimen. This reflect the uncertainty in the management of patients with HFpEF. However, the most recent guidelines recommend the use of hypertension medications such as ACEi and ARBs along with BBs as appropriate (*Heidenreich et al., 2022*). The guidelines also recommend the use of ARNI to decrease hospitalizations, based on the results of the prospective comparison of ARNI with ARB global outcomes in HFpEF (PARAGON-HF) (*Solomon et al., 2019*), yet with lower level of evidence. Moreover, the results of empagliflozin in HFpEF trial (EMPEROR-Preseved) (*Anker et al., 2021*) and the dapagliflozin evaluation to improve the lives of patients with HFpEF trial (DELIVER) (*Solomon et al., 2022*) showed positive results on the use of SGLT2i. The aforementioned trials along with the trials on steroidal (*Pitt et al., 2014*) and non-steroidal MRAs (*Solomon et al., 2024*) highlight that the use of the four-pillar regimen for HF might also be beneficial among patients with HFpEF.

In our study, data about the reasons of not using the four-pillar of HF is not available. SGLT2i is a medication used to manage type 2 diabetes yet, the EMPA-REG OUTCOME trial demonstrated that among patient with type 2 diabetes, SGLT2i was associated with significant reduction in HF hospitalizations (*Anker et al., 2021*). These benefits were persistent even among patients with no history of HF. In addition, these benefits were independent from improvement in renal function, (*Anker et al., 2021*; *McMurray et al., 2019*) glucose levels (*Fitchett, McKnight & Lee, 2017*) and baseline glycated hemoglobin (*Fitchett, McKnight & Lee, 2017*; *Fitchett, Udell & Inzucchi, 2017*). Moreover, patients with both HF and CKD had 25% increase in risk of mortality compared to patients with HF alone (*Ather et al., 2012*). SGLT2i also offer benefit to patients with HF through renoprotection with this effect mediate a portion of the reduction in HF hospitalization observed with SGLT2i (*Lam et al., 2019*). Although the medication also showed great benefits in reducing mortality and hospitalization rates among patients with HFpEF (*Anker et al., 2021*) and HFrEF (*Zinman et al., 2015*), we demonstrate that its utilization is very low in our study.

Several limitations should be acknowledged. Due to the observational design of this study, it is not possible to establish cause-and-effect relationships for many of the associations mentioned. Additionally, we cannot dismiss the potential impact of unmeasured confounding factors on the non-use of the four-pillar of HF. Data about the specific types of BB, ARNI, ACEI, and SGLT2i used in the registry was unavailable

in our dataset. The applicability of our findings to other countries relies on similarities in population characteristics, healthcare systems, and heart failure management. An additional limitation is the short follow up time in our study which limits our findings on mortality and hospitalization rates. However, an ongoing work in our group to collect data on one-year follow up. Additionally, we did not find differences in survival rates between patients on the four-pillar and patients who were not which contradicts previous findings (*D'Amario et al., 2023*). However, we consider our analysis limited due to the low number of patients who were on the four-pillar regimen for HF. Finally, a significant limitation of this and other registry studies is the absence of longitudinal data and the ability to establish the timing of clinical variables, particularly prior measurements of serum potassium levels, which could have influenced decisions regarding the initiation of therapy and medication usage.

In conclusion, the use of the four-pillar regimen for HF was very low in Jordan. Patients with history of type 2 diabetes and impaired renal function, were more likely to receive the four-pillar regimen for HF highlighting theunderuse of the four-pillar regimen. Governmental initiatives should focus on reducing the underuse of these medications by addressing high costs, low availability, and poor insurance coverage.

### Funding
The authors received no funding for this work.

### Competing Interests
The authors declare there are no competing interests.

### Author Contributions
- Mahmoud Izraiq conceived and designed the experiments, performed the experiments, authored or reviewed drafts of the article, and approved the final draft.
- Mustafa Jammal performed the experiments, authored or reviewed drafts of the article, and approved the final draft.
- Ahmad A. Toubasi conceived and designed the experiments, performed the experiments, analyzed the data, prepared figures and/or tables, authored or reviewed drafts of the article, and approved the final draft.
- Sae'ed Mari performed the experiments, authored or reviewed drafts of the article, and approved the final draft.
- Sarah AlNajafi performed the experiments, authored or reviewed drafts of the article, and approved the final draft.
- Ayad Al-Qadasi performed the experiments, authored or reviewed drafts of the article, and approved the final draft.
- Khaled Al Maharmeh performed the experiments, authored or reviewed drafts of the article, and approved the final draft.

- Maha Almansour performed the experiments, authored or reviewed drafts of the article, and approved the final draft.
- Soadad Saleh performed the experiments, authored or reviewed drafts of the article, and approved the final draft.
- Yaman Ahmed performed the experiments, authored or reviewed drafts of the article, and approved the final draft.
- Hadi Abu-hantash conceived and designed the experiments, performed the experiments, authored or reviewed drafts of the article, and approved the final draft.

## Human Ethics

The following information was supplied relating to ethical approvals (i.e., approving body and any reference numbers):

The Institutional Review Board (IRB) at the Specialty Hospital has reviewed and approved the conductance of this study (approval number: 5/1/t/104826).

## Data Availability

The raw data is available in the Supplementary File.

## Supplemental Information

Supplemental information for this article can be found online at http://dx.doi.org/10.7717/peerj.18464#supplemental-information.

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
