# Peer review of "The use of four-pillar regimen for heart failure management: results from the Jordanian Heart Failure Registry (JoHFR)"

_PeerJ, doi:10.7717/peerj.18464_

## Round 0.1 · original submission · Major Revisions

· Academic Editor

Major Revisions

Thank you for submitting your manuscript titled The use of the 4 pillars of HF management and the factors associated with it: results from the Jordanian Heart Failure Registry (JoHFR)". The reviewers have provided thorough and constructive feedback, particularly Reviewer #1, whose detailed review (in their PDF) offers several key suggestions for improving the manuscript.

The manuscript would benefit from a refinement in writing style to improve clarity and readability. Reviewer #1 noted unnecessary wording that could be eliminated to streamline the text. Additionally, it is recommended not to use abbreviations or acronyms, such as "HF" for "Heart Failure," in the title. This will ensure the title is clear and accessible to a broader audience.

The presentation of the results does not flow smoothly, and crucial outcome results are missing. Specifically, the inclusion of patients with Heart Failure with preserved Ejection Fraction (HFpEF) raises concerns because the four-pillar heart failure management strategy, including quadruple therapy, is not typically indicated for HFpEF, except for SGLT2 inhibitors. This point needs to be addressed in the results and discussion sections.

The study should clarify whether there were any documented reasons that prevented patients from receiving heart failure medications, such as issues related to renal function or low blood pressure. This information is crucial for understanding the context of the treatment decisions and should be discussed in the manuscript. Additionally, while the study mentions a one-year follow-up, it only reports 30-day mortality. Hospitalization is a key outcome in heart failure research and should be included in the results.

Reviewer #1 pointed out the absence of medication doses in the data collection, which could provide valuable insights into treatment adequacy and patient outcomes. Moreover, the manuscript should address the accuracy and completeness of the registry data and report any missing data, which are inherent limitations of retrospective studies.

According to Reviewer#2 the literature review should be expanded to include recent guidelines from the American College of Cardiology (ACC), American Heart Association (AHA), Heart Failure Association (HFA), and European Society of Cardiology (ESC). Additionally, the discussion should incorporate the role of heart failure medications used for Type 2 Diabetes Mellitus (T2DM) and the impact of newer drugs on patients with decreased estimated Glomerular Filtration Rate (eGFR).

Reviewer #1 suggested adding a Kaplan-Meier (KM) curve for outcomes, which would enhance the presentation of the results. This visual representation could provide a clearer understanding of the survival analysis over the follow-up period.

Once these revisions are made, we would be happy to reconsider your manuscript for publication.

·

Basic reporting

It is recommended to follow the STROBE checklist for reporting observational studies. English style needs to be refined and get rid of unnecessary working. More details in the attached documents.

Experimental design

Many essential aspects of the methodology need to be clearly defined. More details in the attached documents.

Validity of the findings

More details in the attached documents.

Additional comments

More details in the attached documents.

·

Basic reporting

- English language need to improve and sentence structuring need improvement
- Literature review is inadequate
- Figures are Ok, but need to add KM curve for outcomes
- Need to discuss what authors purpose how to improve the GDMT in HF patients

Experimental design

- Retrospective study

Validity of the findings

-Retrospective registry data with inherent limitation
- Should report missing data
- How accurate and completeness of the data

Additional comments

Authors should add to their discussion and citations
-most recent guidelines from ACC/AHA/HFA and ESC guidelines
-HF medications used for T2DM
- Discuss the role of newer drugs and decreased eGFR

---

## Round 0.2 · Minor Revisions

· Academic Editor

Minor Revisions

Thank you for submitting the revised version of the manuscript and for the work done so far. After a careful review of this new version, we have identified that, while progress has been made in many areas, there are still some points that can be improved to ensure that the work fully meets the quality and rigor standards expected by our journal.

We kindly ask that you consider these comments and revise the manuscript accordingly. We look forward to receiving the new version with these improvements.

Please do not hesitate to reach out if you have any questions or require further clarification.

·

Basic reporting

Kindly check comments on the attached document in the bubble notes and as highlighted in the yellow color.

Experimental design

Kindly check comments on the attached document in the bubble notes and as highlighted in the yellow color.

Validity of the findings

Kindly check comments on the attached document in the bubble notes and as highlighted in the yellow color.

Additional comments

Kindly check comments on the attached document in the bubble notes and as highlighted in the yellow color.

There was a good improvement in the writing style but a bit more will make the paper much better.

·

Basic reporting

This is review of revised manuscript. The authors have answered all my queries. I have no further comments.

Experimental design

This is review of revised manuscript. The authors have answered all my queries. I have no further comments.

Validity of the findings

This is review of revised manuscript. The authors have answered all my queries. I have no further comments.

Additional comments

None

---

## Round 0.3 · accepted · Accept

· Academic Editor

Accept

Thank you for submitting the revised version of your manuscript. After carefully reviewing the changes, I can confirm that all reviewers' comments and suggestions have been appropriately addressed.

Based on this assessment, I am pleased to inform you that the manuscript is now ready for publication. Congratulations.